# A Deep Learning Approach to Estimate the Incidence of Infectious Disease Cases for Routinely Collected Ambulatory Records: The Example of Varicella-Zoster

**DOI:** 10.3390/ijerph19105959

**Published:** 2022-05-13

**Authors:** Corrado Lanera, Ileana Baldi, Andrea Francavilla, Elisa Barbieri, Lara Tramontan, Antonio Scamarcia, Luigi Cantarutti, Carlo Giaquinto, Dario Gregori

**Affiliations:** 1Unit of Biostatistics, Epidemiology and Public Health, Department of Cardiac, Thoracic, Vascular Sciences and Public Health, University of Padova, Via Loredan, 18, 35131 Padova, Italy; corrado.lanera@unipd.it (C.L.); ileana.baldi@unipd.it (I.B.); andrea.francavilla@ubep.unipd.it (A.F.); 2Division of Pediatric Infectious Diseases, Department of Women’s and Children’s Health, University of Padova, 35131 Padova, Italy; elisa.barbieri@unipd.it (E.B.); carlo.giaquinto@unipd.it (C.G.); 3Consorzio Arsenàl.IT, 35131 Padova, Italy; ltramontan@consorzioarsenal.it; 4Società Servizi Telematici–Pedianet, 35138 Padova, Italy; a.scamarcia@sosepe.com (A.S.); l.cantarutti@sosepe.com (L.C.)

**Keywords:** electronic health records, infectious disease, varicella-zoster, deep learning, natural language processing

## Abstract

The burden of infectious diseases is crucial for both epidemiological surveillance and prompt public health response. A variety of data, including textual sources, can be fruitfully exploited. Dealing with unstructured data necessitates the use of methods for automatic data-driven variable construction and machine learning techniques (MLT) show promising results. In this framework, varicella-zoster virus (VZV) infection was chosen to perform an automatic case identification with MLT. Pedianet, an Italian pediatric primary care database, was used to train a series of models to identify whether a child was diagnosed with VZV infection between 2004 and 2014 in the Veneto region, starting from free text fields. Given the nature of the task, a recurrent neural network (RNN) with bidirectional gated recurrent units (GRUs) was chosen; the same models were then used to predict the children’s status for the following years. A gold standard produced by manual extraction for the same interval was available for comparison. RNN-GRU improved its performance over time, reaching the maximum value of area under the ROC curve (AUC-ROC) of 95.30% at the end of the period. The absolute bias in estimates of VZV infection was below 1.5% in the last five years analyzed. The findings in this study could assist the large-scale use of EHRs for clinical outcome predictive modeling and help establish high-performance systems in other medical domains.

## 1. Introduction

Disease burden estimates for infectious diseases are crucial for public health resource allocation [1].

Active and passive surveillance systems are essential to evaluate the epidemiological impact of infectious diseases: they can provide background data to implement effective control strategies, such as vaccination campaigns, and monitor the trend [2].

In this framework, the detection of varicella-zoster virus (VZV) has recently raised interest due to the extensive debate about the need and cost–benefit profile of introducing a mass vaccination program for young people [3,4]. The national routine notification system adopted in Italy—which is mandatory—helps describe the epidemiology of VZV and allows one to evaluate historical temporal trends but is undoubtedly affected by under-notification and under-diagnosis [5].

The broad implementation of the Electronic Health Record (EHR) in primary care offers new opportunities for population-based studies through data generated by accesses in any care setting, making it accessible promptly [6,7]. In addition, automatic methods for data extraction from free-text reports are increasingly replacing manual extraction [8,9]. In particular, machine learning (ML) is gaining popularity in healthcare because of its power to extract and filter available information from big data and accurately solve demanding learning tasks such as classification, clustering, and numerical prediction [10,11].

In recent years, enhanced generalized linear model (GLM) techniques have been used for text mining from EHRs for the case detection of VZV [12]. However, these techniques can produce highly variable resulting models depending on the analyst’s choices, especially in a class imbalance [13].

Standard shallow ML algorithms cannot directly model data sequences from text streams: they rely on the independence assumptions between tokens, i.e., a single unit of textual information, which generally corresponds to “words,” covariates. Deep-recurrent neural network architectures can overcome these limitations: the reduction of input preprocessing and manipulation, the possibility to process the text as a data sequence, and the automatical learning of the correlations between features without superimposed structures meet the mentioned needs [14,15].

Applying the mentioned approach could, in turn, be used to achieve two critical goals: to estimate the incidence of VZV infection by automatic case identification and to make timely and consistent predictions over time.

### Highlights

Estimates of infectious disease incidence can be time-consuming and tedious;Deep learning, in particular RNN-GRU, for automatic data extraction from the free text could be a feasible and timely option;Results obtained with MLT were promising, yet, in future development, this text-mining tool should be readily usable by non-technical users as well.

## 2. Materials and Methods

### 2.1. Electronic Medical Record Database

The Pedianet database (http://www.pedianet.it/en/, accessed on 9 May 2022) is a pediatric primary care database that contains clinical, demographic, prescription, and outcome data of children aged 0–14 years. Data are generated during daily clinical practice by about 150 family pediatricians who use the same software JuniorBit^®^ (various versions); in the recent past, it has been exploited for infectious disease research studies [12,16,17]. In addition, Pedianet gathers details about specialist referrals, procedures, hospitalizations, medical examinations, and health status (according to the Guidelines of Health Supervision of the American Academy of Pediatrics).

Pedianet was the starting point for producing the gold-standard diagnosis of VZV infection for each record, according to the literature [18].

The study population includes all the children in the Veneto region (Northeast Italy) who were visited at least once between 2004 and 2014: data about 1,227,578 visits and 7631 children were collected. The baseline characteristics are shown in Table 1. stratified for outcome class, i.e., negative or positive case of VZV in the corresponding year. Sex, a categorical measure, is reported as a percentage and absolute frequency. Age, a continuous measure, is reported with I/II (median)/III quartiles.

### 2.2. Main Strategy

VZV infection exhibits unique characteristics. First, it can be contracted once in a lifetime (with notable exceptions). Furthermore, using a specific diagnostic code to identify VZV infection is not mandatory for a pediatrician. Consequently, it can be critical to ascertain cases and estimate the yearly incidence rate of VZV infection.

Three Pedianet researchers spent two years manually reviewing the medical records to obtain a gold standard for VZV infection; due to this mentioned period, a delay of two years was set as the starting point to develop our model. The main idea was to train the model to detect—through natural language processing (NLP)—the status of a child, i.e., infected or not by VZV, based on text fields of the Pedianet, to compare the predicted incidence with the gold standard and to use the newly acquired data to improve predictive performances and continuously update the existing models.

For illustrative purposes, let us assume that data for a given year are available, e.g., 2006. Under a “two-year delay,” the gold standard is established for data up to 2004. A first version of the model is trained on data up to 2004 and applied to classify data from 2005 to 2006. The following year, i.e., 2007, the model can be updated with the data from 2005 (only in 2007, the gold standard for data from 2005 is supposed to be available) and used to classify all the cases collected in the last two years, i.e., 2006 and 2007. This process went on year by year up to the last model trained on data from 2004 to 2013 and used to classify 2014 data (i.e., the last year for which we have the gold standard). Overall, 10 models were trained, i.e., with training data from 2004 alone, 2004–2005, 2004–2006, 2004–2007, 2004–2008, 2004–2009, 2004–2010, 2004–2011, 2004–2012, 2004–2013, each of them used to classify and to be tested on the next two years, i.e., 2005–2006, 2006–2007, 2007–2008, 2008–2009, 2009–2010, 2010–2011, 2011–2012, 2012–2013, 2013–2014, and 2014 alone, respectively. The general strategy is described in Figure 1.

The entire dataset was initially divided into ten groups, indexed by the last year of training data, from 2004 to 2013. Each group has two components. The first component contains the training data, divided into 80% and 20% for training and validation/fine-tuning, respectively. The second component includes the training data (from 2004 to the reference year) and the next two years of data for testing purposes. Models were tuned on the first component of each group, and hyperparameters were selected for optimizing each validation set. Once the tuning parameters were selected, the new models were (re-) trained on the whole training set using that selection of parameters, and tested on the testing one, i.e., the next two years respect the last one used in the corresponding training set. See Table 2 for the sets’ sizes.

### 2.3. Model Choice

Given the sequential nature of the input for the task, a deep learning architecture involving a recurrent component to process the text was set up. Compared to a traditional neural network, a recurrent neural network (RNN) is specifically designed to grasp the complex relations between sequential data, such as text reported in natural language.

Within the RNN base components, the gated recurrent unit (GRU) was chosen to meet the needs. A GRU can be viewed as an optimized Long Short-Term Memory (LSTM) module, i.e., the principal reference for advanced RNN networks [19]. LSTM, as well as its recent customized versions, would have also represented a valid option to explore [20]. On the other hand, we preferred GRU to LSTM and its variations because GRU has lower complexity and required resources [21]. In particular, GRU meets the needs of a fast-converging learner who can make an accurate prediction within a reasonable amount of computational resources and time.

### 2.4. Language Model

The text must be converted into numbers to serve as input in an ML model. One of the most effective strategies to represent the text as a vector of numbers is to resort to embeddings. Vectors are “trained” to convert syntactically and semantically similar words into geometrically close points of a vector space. Due to the highly specialized field and the high amount of data disposal, a self-trained embedding representation was chosen. The embeddings were created using the FastText algorithm with the SkipGram strategy and 300-dimension output vectors [22]. The entire Pedianet database was used, with an overall of 6,903,035 visits of 216,976 children collected by 144 family pediatricians starting from 1 January 2004 to 23 August 2017.

### 2.5. Implementation

First, the records were collapsed and arranged in chronological order to allow the GRU to work correctly. In the Pedianet, each row represented all available data for a specific visit of a child. Next, all the records for a given year were put in a single text cell. Each row was finally composed of a single cell reporting all the sequentially ordered history of textual notes for a child in a specific year.

Furthermore, it was necessary to have a fixed dimension for the input sequence to allow bidirectional RNN: a hard limit of 10,000 words was chosen. Records with more than 10,000 words were truncated (loss of information), while those with less than 10,000 words were filled with a fictitious word (__PAD__).

After these operations, each input was a three-dimensional matrix with rows corresponding to the corresponding amount of record (see Table 2), 10,000 columns (words in each record), and 300 deep-wise elements (embedding representation for each word).

A bidirectional RNN-GRU was added to the network considering 256 nodes in each direction (512 in total).

All the processed information was analyzed by two fully connected layers of 128 nodes each (256). The last 128 nodes are next connected to a single node with a logistic activation to output the probability of the possible status: VZV positive versus VZV negative for every child.

To summarize, the overall network was composed of the following main layers (see Figure 2):
Embeddings: representation of words converted into a syntactic and semantically coherent 300-dimensional structure—input N × 10.000/output N × 10.000 × 300, where N is the number of cases (i.e., one case is the collection of all the HERs of a single child for a given year) in the dataset/minibatch considered.Two synchronized layers of bidirectional RNN-GRU modules are composed of 256 nodes each (512 nodes of each layer process the information as a pure sequence summarizing its “meaning”—input N × 10.000 × 300/output N × 512 (output for the first synchronized layer is equal to the input of the second: N × 10.000 × 512))Two fully connected layers of 128 nodes each to process the “meaning” vector of information from each record—input N × 512/output N × 128 (output for the first synchronized layer is equal to the input of the second: N × 128).Logistic output node: to produce a probability measure for the children being affected by VZV in the corresponding year—input N × 128/output N × 1.
Figure 2Flowchart of the trained network. The boxes report shapes and shape interpretation of the data between each computation step, i.e., between layers of the network. Layers are reported as linking connections between the boxes. N represents the size of the record passed in input; for our minibatch training, N is 16; for overall records, N depends on the set reported in Table 2.
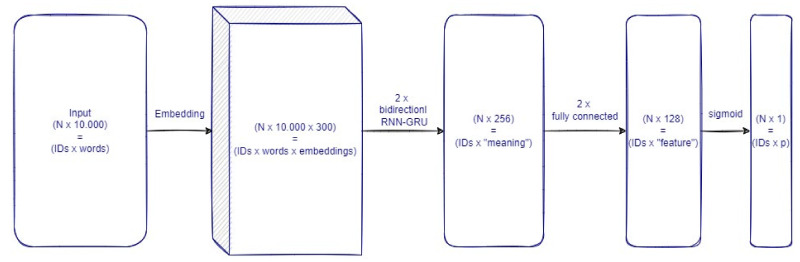


### 2.6. Training

The training of the network was based on the optimization/minimization of the weighted binary cross-entropy log-loss (wbcell) measure, i.e., (minus) the mean of the logarithm of the probability of being classified in the correct class (i.e., $p_{i}$ for positive cases, $(1-p_{i}$) for the negative ones). For taking into account the issue of class imbalance, the contributions to the loss estimation were weighted by the inverse of the class relative frequency (considering $f_{P}$ the absolute frequency of positive cases, and $f_{N}$ the absolute frequency of the negative ones). With the notation adopted the optimization measure takes the following expression:$$wbcell=\frac{1}{N}\overset{N}{\underset{i=1}{\sum }}-(\frac{N}{f_{P}}*y_{i}*\log\left(p_{i}\right)+\frac{N}{f_{N}}*\left(1-y_{i}\right)*\log\left(1-p_{i}\right))$$

For network optimization, the Adam strategy was adopted [23]. The network was fed with a mini-batch of size 16, i.e., 16 records at a time [24]. An “epoch” was passed once the network was trained by all the records (batch by batch). The number of training loops was capped at 15 epochs, imposing an early stop if no improvement was seen within 5 epochs and choosing the best epoch [25].

A small nodal dropout was performed across the whole network to minimize the over-fitting [26].

The output model is a probabilistic one, i.e., for any given case, the output is its estimated probability of being a positive one. Deciding if a case is positive or not, i.e., defining a classifier, is a matter of selecting a cut-off for those probabilities. Selecting the cut-off is mainly a matter of utility/preference in preferring false negative or false positive as a potential error [27]. To evaluate a model independently from the cut-off, we can draw the receiver operating characteristic curve, i.e., the curve made up of all the points of coordinate ($sensitivity_{c}$, $1-specificity_{c}$), at any possible cut-off c, from 0 to 1. The more area under the ROC curve, the better the model is. The area under the ROC curve (AUC-ROC) is used as the evaluation metric for the model trained. AUC-ROC ranges from 0 to 1, with a 0.5 value representing uninformative models, i.e., equivalent to random guesses.

## 3. Results

In total, 1,227,578 records recorded on Pedianet databases from 7631 children between 2004 and 2014 were merged by patient and year. They were divided into ten cumulative training sets, starting from data from 2004 and adding a year each time. Correspondingly, 10 test sets containing the two years’ data following the last one in the training set were built up. A dictionary of 122,607 words was trained as 300-dimension embedding vectors from Pedianet textual entries.

Across the ten model years, a median AUC-ROC of 97.97% (IQR 97.25%–98.07%) within the training sets and 71.36% (IQR 60.35%–82.47%) for the test sets were reached. A maximum test AUC-ROC of 95.30% was reached for the 2013 model, i.e., the last and the one with the most data in the training set. The minimum test AUC-ROC of 38.46% was reached for the model with data up to 2005. Progression of AUC-ROC performances in the training and test set are reported in Figure 3.

The performance of the training sets increases quickly in the initial models and then stabilizes. On the contrary, the performance of the test sets continuously improves over time, suggesting that the underlying model is suitable for the specific task (optimal training performances) and benefits from the update with new data to generalize its usability (increasing testing performances).

The outcomes of the trained models convey a probability for each case to be a positive one in the corresponding year; before obtaining a classifier, a threshold cut-off for the probabilities had to be selected to distinguish between positivity and negativity. Thus, an optimal cut-off was chosen for maximizing the product of precision (i.e., the portion of records correctly classified as positives) and recall (i.e., the portion of positive records that are correctly classified) [28]. The performances corresponding to each test set for all the models are reported in Table 3.

Each model, equipped with the corresponding cut-off, is then used to estimate the incidence of VZV infections in the two following years of the test. Results are reported in Table 4.

ROCs curves, highlighting the cut-off point and the corresponding difference in the estimation of VZV infections, are reported in Figure 4. The years of the training models are reported on the facets’ headers. Testing years are the following two up to 2014. Color variations in the curves represent the variation of the error in the incidence estimation. The optimal cut-off maximizing the product of precision and recall is reported (red dot) on the side of the corresponding error produced by classifying records using it.

### Computational Environment

All the computations were performed using the R v4.1.2 [29] programming language, powered by TensorFlow and compiled for GPU usage, and its Keras interface for the deep learning modeling and training, the {targets} R-package for the global pipeline control, and the {tidyverse} R suite of packages for data management and plots. All the computations run on a Linux Ubuntu 20.04 operating system installed on a machine with a 16-Core Xenon processor equipped with 128 GB RAM and a CUDA NVIDIA Quadro RTX5000 graphic card with 16 GB of dedicated memory.

## 4. Discussion

Many medical organizations and networks still use manual processes to extract data from unstructured EHRs. Especially for infectious diseases, delays in producing and disseminating the results may hinder a prompt public health response. Thus, it is necessary to create and continuously update a data-based infectious disease prediction model to handle situations in real time [21].

While automatization, in general, represents a hot topic nowadays, the potential of these approaches still has to be fully explored, and the state of the art is hard to identified [30]. On the other hand, some recent works have already explored the field, finding potential best practices to address the issues concerning the classification of natural free-text fields from EHRs. Accordingly to our first step, one of the strategies appearing promising is the adoption of pre-trained language models as a starting macrolayer of the network [31].

This work proposes an alternative strategy to set up, maintain, and update a data-based VZV surveillance model to supplement existing systems. It could be thought of as the starting point to solving the issues within the medical database. It would be interesting to know whether NLP—a very well-known topic for other application fields—could be successfully used in this context [32]. It opens new options and perspectives to analyze medical data in an integrated environment.

Real-incidence rates would be at disposal when manual extraction is performed. Conversely, the automated extraction represented by the model purposed could provide instant estimates for the previous years: e.g., on the 1st of January of a specific year, when all the data about the previous year were collected, it could be possible to estimate the yearly incidence for a specific pathology. Moreover, it could also be possible to have an updated estimation of the number of cases detected up to now live. Finally, thanks to the NLP models, it is possible to calculate incidence for the upcoming years as well—i.e., by making predictions—with better precision than simply relying on recent incidence trends.

Several studies have used techniques from deep learning to predict infectious diseases. Our findings align with those of studies reporting that deep learning yields satisfactory results when used to perform tasks that are difficult for conventional analysis methods [33]. For example, through an indirect comparison of the same Pedianet data [12], RNN-GRU yielded better VZV prediction performance than enhanced GLM-based ML models. As such, methods for predicting infectious diseases, such as VZV that uses deep learning, help design effective models.

In future scenarios, the framework mentioned above could be expanded and used in similar tasks of predicting other infections than VZV ones. None of the methodologies are specific for VZV, and the whole procedure can be specialized and fine-tuned for other targets. One of the main advantages of (recurrent) neural networks is their ability to keep themselves up to date with low effort, thanks to the pre-trained scenario for transfer learning. Much of the ability learned by the network is retained, and only the terminal nodes used for the final classification are fine-tuned for the specific target of interest [34]. Due to this reason, the strategy is getting popular even for the general-purpose classification of biomedical text [35].

Furthermore, in the present work, the models learn from free-text fields only, without any additional input. Any evidence or suggestions from the literature were not given to our model; it is well known that some factors influence the VZV incidence. For example, the peak of infections is in the preschooler age and decreases afterwards. Given these considerations, our work could be thought of as part of a more complex data integration system, where essential variables of different nature and importance are put together to help maximize the desired result [36]. Deep learning is especially suitable and able to manage by design this mixed-type of information, from structured (e.g., tabular information) to unstructured types (e.g., free-text). Both the scenarios are promising: the former, where structured clinical data were used as additional metadata to improve NLP tasks, and the latter when NLP produces small, structured summary information to integrate classical statistical models.

### Limitations

#### Some Limitations Must Be Acknowledged

The amount of textual-based information stored electronically is rapidly increasing. Accumulating information is easy; however, finding relevant information on demand can be difficult as the size of the collection continues to escalate. This article presents a general framework for text mining consisting of a large body of EHRs with text data that are inherently unstructured and fuzzy and cutting-edge advances in deep learning. However, barriers to adoption are one of the main challenges. Trusting that the current system extracts high-quality information (i.e., all actual VZV cases) is also likely to cause concern that an approach that deviates from the current standard (i.e., manual review) might not be of equal quality. In addition, this system is designed for trained knowledge specialists. In future developments, this text-mining tool should be readily usable by non-technical users.

It was recently shown that the Bidirectional Encoder Representations from Transformers (BERT)-based model outperforms other deep learning alternatives in classification tasks from EHRs [37]. BERT is based on a bi-directional representation of tokens incorporating attention layers. Pre-train BERT language models require a massive amount of computational resources (in contrast with FastText, i.e., our choice). In contrast, using a pre-trained one is significantly lighter. On the other hand, in the present work, we preferred to start exploring potential solutions based on a personalized pre-trained language model. For that reason, we did not adopt a BERT-first approach, leaving that for future exploration.

While the architecture does not theoretically require GPUs to be trained or evaluated, it is complex and inefficient for a general-purpose production environment. Exploration of optimization strategies for lowering the computational complexity of the network evaluation can be considered for real-world usage.

While the AUC-ROC on the training set goes up quite quickly (basically after the first year of data), AUC-ROC on the test set needs more time. It obtains satisfying results only when data were available for nine years and passing the 0.9 area only when ten years were considered. Of course, this variability reflects the dimension of the starting dataset.

It is also worth considering the actual nature of the task in supporting human efforts and decisions. In the present work, we have “mathematically” optimized the classifiers derived from the probabilistic models selecting a cut-off and optimizing the product of precision and recall. Given the high level of AUC-ROC reached by the final optimized models, the cut-off can be selected to improve the recall up to a trade-off that allows a human reviewer to only inspect a (small amount of) positive predicted record to exclude false-positive detection. That way, a dual benefit can be obtained: a more precise estimation of the incidence with a low additional human effort and a selection of highly relevant wrongly classified records that can be specifically used to further refine the model in the next update.

It is essential to understand that our model is not temporal; thus, it aims to obtain the case identification and benefit from additional data, year after year, rather than find a correlation in a specified temporal space. Its results are likely to help integrate temporal models on the side of other structured data.

There is also a lack of external validation. Other pediatricians in different regions with different habits can challenge the model. That is reflected in the high performance obtained and maintained on the training set since the early phases, while more time and data are needed to reach a similar level of performance on the testing sets. For training a deep learning NLP classifier, a large amount of different data are needed to explore the variability of language used in the field. On the other hand, as our study shows, with a sufficient amount of data, the model can learn enough to be used on entirely new data with similar performances shown during its training phase.

## 5. Conclusions

The proposed method achieves promising results which outperform other state-of-the-art algorithms, showing that the NPL approach can effectively predict the status of a child with reasonable accuracy. The deep-learning-based system built in this study could be applied to facilitate the large-scale use of family pediatrician notes for clinical outcome predictive modeling. The findings in this study could also assist in establishing high-performance systems in other medical domains integrating structured and unstructured data.

## Figures and Tables

**Figure 1 ijerph-19-05959-f001:**
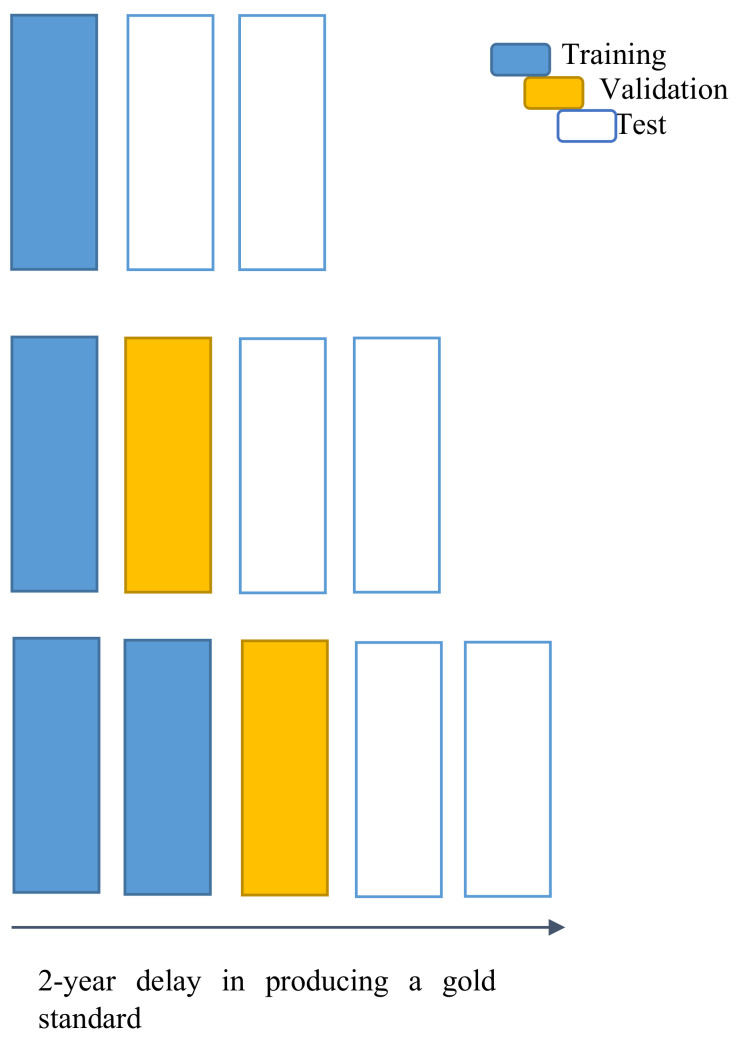
Chart for the general strategy for model development and test. Top-line on the chart: each new year X (right-most position on the x-axis), an updated model can be trained on the already ready gold-standard data, i.e., up to the previous two years (blue), and used to predict the following two years X − 1 and X (white). Middle-line on the chart: the following year X + 1, a second, updated prediction (yellow) can be made on one of the years of tested data (X − 1) with the previous model. Bottom-line on the chart: in the second next year X + 2 the gold standard is supposed to be ready for that year X − 1, becoming a new training data (blue). The model can provide an updated prediction for the year X and a new prediction for the years X + 1 and X + 2 (i.e., the current one “just ended”).

**Figure 3 ijerph-19-05959-f003:**
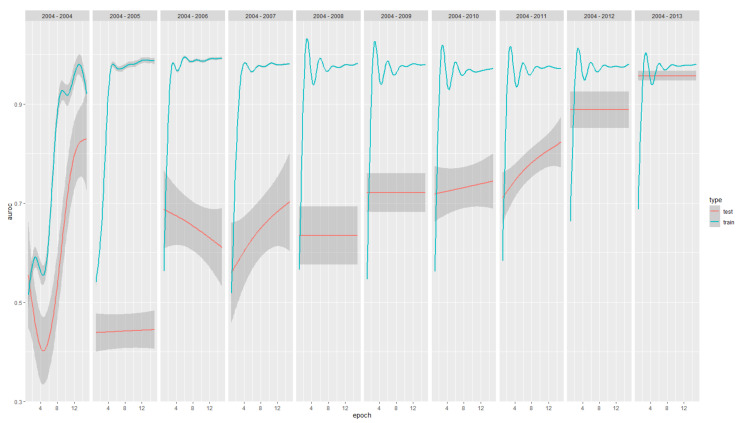
AUC-ROC (y-axis) performance progression across epochs of training (x-axes) and model years(panels from 2004 to 2004–2013, from left to right) for both the train (green) and test (red). In total, 95% CI are reported as shadows.

**Figure 4 ijerph-19-05959-f004:**
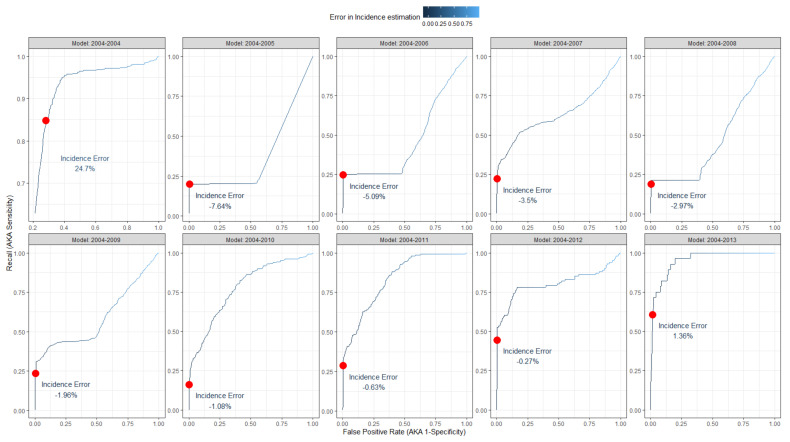
Receiver operator curves (ROCs) of the model trained to classify VZV infections. The years of the training models are reported on the facets’ headers. Testing years are the following two up to 2014. Color variations in the curves represent the variation of the error in the incidence estimation. The optimal cut-off maximizing the product of precision and recall is reported (red dot) on the side of the corresponding error produced by classifying records using it.

**Table 1 ijerph-19-05959-t001:** Characteristics of units analyzed. Descriptive statistics are reported and stratified for outcome class, i.e., negative or positive case of VZV in the corresponding year.

	N	VZV Negative (N = 58,334)	VZV Positive (N = 2325)
Sex	60,659		
Female		47% (27,340)	46% (1068)
Male		53% (30,994)	54% (1257)
Age [days]	60,342	0.7/2.2/4.28 *	0.6/1.4/3.1

* I/II (median)/III quartile.

**Table 2 ijerph-19-05959-t002:** Cases in each set of the models trained. All child records for a given year represent a case, i.e., the same child in distinct years represents distinct and independent cases. Each row reports datasets for the training, validation, and test of a model.

Years		Training Phase		Testing Phase	
Train	Test	Train (#)	Validation (#)	Train (#)	Test (#)
2004	2005–2006	1588	396	1984	7854
2004–2005	2006–2007	4405	1099	5504	9454
2004–2006	2007–2008	7873	1965	9838	10,852
2004–2007	2008–2009	11,969	2389	14,958	12,020
2004–2008	2009–2010	16,555	4135	20,690	13,062
2004–2009	2010–2011	21,586	5392	26,987	13,848
2004–2010	2011–2012	27,006	6746	33,752	14,139
2004–2011	2012–2013	32,666	8160	40,826	14,017
2004–2012	2013–2014	38,319	9572	47,891	12,768
2004–2013	2014	43,882	10,961	54,843	5816

**Table 3 ijerph-19-05959-t003:** Number of positives, negatives, area under the receiver operating characteristic curve (AUC-ROC), predicted true-positives (tp) and true-negatives (tn), false-positives (fp), and false-negatives (fn). Precision or positive predictive value (prec) and recall or sensitivity (rec) for each model year (by row, indexed by the column year) related to their corresponding test sets. Bold face is used to highlight the best performance column wise.

Model Year	Positives	Negatives	AUC ROC	tp	tn	fp	fn	prec	rec
2004–2004	637	1.954	0.804	540	5.180	2.037	97	0.210	0.848
2004–2005	172	3.348	0.385	188	8.474	35	757	0.843	0.199
2004–2006	465	3.869	0.588	194	10.024	41	593	**0.826**	0.247
2004–2007	480	4.640	0.649	130	11.403	33	454	0.798	0.223
2004–2008	307	5.425	0.582	102	12.470	51	439	0.667	0.189
2004–2009	277	6.011	0.652	98	13.386	46	318	0.681	0.236
2004–2010	264	6.510	0.775	37	13.870	40	192	0.481	0.162
2004–2011	152	6.922	0.835	43	13.848	19	107	0.694	0.287
2004–2012	77	6.988	0.832	45	12.645	22	56	0.672	0.446
2004–2013	73	6.879	**0.953**	17	5.698	90	11	0.159	**0.607**

**Table 4 ijerph-19-05959-t004:** Incidences of VZV infections observed in Pedianet and estimated by the model trained.

Model Year	Years Estimated	Positives	Negatives	Observed Incidence (%)	Estimated Incidence (%)	Estimated Incidence Error (%)
2004	2005–2006	637	1.954	8.11	32.8	24.7
2004–2005	2006–2007	172	3.348	10	2.36	−7.64
2004–2006	2007–2008	465	3.869	7.25	2.17	−5.09
2004–2007	2008–2009	480	4.640	4.86	1.36	−3.5
2004–2008	2009–2010	307	5.425	4.14	1.17	−2.97
2004–2009	2010–2011	277	6.011	3	1.04	−1.96
2004–2010	2011–2012	264	6.510	1.62	0.54	−1.08
2004–2011	2012–2013	152	6.922	1.07	0.44	−0.63
2004–2012	2013–2014	77	6.988	0.79	0.52	−0.27
2004–2013	2014	73	6.879	0.48	1.84	1.36

## Data Availability

Data available on reasonable request from Pedianet. Code developed is publicly available at www.github.com/UBESP-DCTV/varicella.due (accessed on 9 May 2022).

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
