# Peer review of "A Deep Learning Approach to Estimate the Incidence of Infectious Disease Cases for Routinely Collected Ambulatory Records: The Example of Varicella-Zoster"

_ijerph, 2022, doi:10.3390/ijerph19105959_

Round 1

Reviewer 1 Report

This study classifies VZV infection by optimizing the parameters of recurrent deep learning net- works while considering a large set of EHRs, spanning over ten years from the Italian PEDIANET 28 database. A bidirectional gated recurrent unit (GRU), a specialized recurrent neural network (RNN), was used to identify varicella cases. 

Some observations:

  1. Highlight the contributions of the proposed work in points under the introduction section.
  2. What are the challenges and limitations of the proposed work?
  3. There is no comparative analysis of proposed work with state of art.
  4. Title in portal and manuscript is different.
  5. What evaluation parameters have been used.

Author Response

  1. Highlight the contributions of the proposed work in points under the introduction section. Thanks for the suggestions. A dedicated paragraph with relevant data is now present in the manuscript.
  2. What are the challenges and limitations of the proposed work? We thank the reviewer for the suggestion; we expanded the Limitation section accordingly.
  3. There is no comparative analysis of proposed work with state of art. The reviewer is correct. We made an indirect comparison with our research group's previous work, where several GLM enhanced techniques were used for VZV case identification from the Pedianet database. Those techniques delivered lower predictive capability than the deep learning approach adopted here.
  4. Title in portal and manuscript is different. We thank the reviewer for pointing out this. The final title is the one reported in the submitted paper. We cannot directly change the portal one. We ask the editor to change the letter accordingly.
  5. What evaluation parameters have been used. Table 3 shows the metrics used for the evaluation of each model: the number of positives and negatives, the area under the receiver operating curve (AUC-ROC), predicted true-positives (tp), and true-negatives (tn), false-positives (fp), and false-negatives (fn). Precision or positive predictive value (prec), and recall or sensitivity (rec). In this regard, additional details are added at the end of the method section.

Reviewer 2 Report

This paper studied a deep learning approach to estimate the incidence of infectious disease cases for routinely collected ambulatory record. Numerical predictions were conducted to verify the feasibility of the proposed method. The reviewer is impressed with the amount of work that went into this research. That alone gives one hope that a paper worthy of the archival standing of a major academic journal such as this is possible. The paper cannot be accepted in the present form as it needs further improvements.  

  1. Abstract: The text must be carefully revised. Some sentences contain mistakes. In a research paper, it is expected that the introduction section briefly explains the starting background and, even more important, the originality (novelty) and relevancy of the study is well established. Once this is done, the hypothesis and objectives of the study need to be addressed, as well as a brief justification of the conducted methodology.
  2. The introduction part does not have a flow or direction. It has too many different medical terminologies thrown randomly. Proper references need to be used rather than using others. Language can be improved. The sentences are half-constructed or incomplete so that the readers are expected to fend for themselves to understand their meaning.
  3. Author must be enriching the references with the latest developments in the field. Some of the recent references can be added. The authors have not paid attention to previous research papers and concerns.
  4. The innovation contribution of this article is not clearly stated. The research contributions should be highlighted in the revised manuscript. There is a certain lack of a clear line and message, and my strong advice to the authors would be to consider the overall structure and to either significantly shorten the manuscript.
  5. Please explain and define all the variables in the equations and check the manuscript thoroughly and define the variables where necessary, otherwise, readers cannot understand the equations.
  6. There are many linguistic and grammatical typos. please carefully read through and conduct the proofreading.
  • Line 31, 44, 55, 118, 197-198, 210-211, 213, 258-259: Your sentence may be unclear or hard to follow. Consider rephrasing.
  1. At the end of the manuscript, please describe the scheme of the intended application of the developed method in real practice. What conditions must be met? What preliminary analysis should be carried out? What is the expected performance of this method? What are the limitations of this method?
  2. Conclusions Section: Improve the conclusions section, it is very general and does not clearly explain the main objectives achieved in this research.

The list could go on, but the bottom line is that the authors need to rewrite the paper or even reconsider the research content before it could be considered for publication in this journal. 

Author Response

  1. Abstract: The text must be carefully revised. Some sentences contain mistakes. In a research paper, it is expected that the introduction section briefly explains the starting background and, even more important, the originality (novelty) and relevancy of the study is well established. Once this is done, the hypothesis and objectives of the study need to be addressed, as well as a brief justification of the conducted methodology. Thanks for the critique. The abstract was re-written according to the reviewer's suggestions.
  2. The introduction part does not have a flow or direction. It has too many different medical terminologies thrown randomly. Proper references need to be used rather than using others. Language can be improved. The sentences are half-constructed or incomplete so that the readers are expected to fend for themselves to understand their meaning. We thank the reviewer for the suggestions. The introduction now incorporates all the advice.
  3. Author must be enriching the references with the latest developments in the field. Some of the recent references can be added. The authors have not paid attention to previous research papers and concerns. We thank the reviewer for the suggestion. We added some recent (2021) references about similar tasks and methods in the methodology, discussion, and limitations sections, highlighting our approach's strengths, weaknesses, and choices concerning alternative state-of-art models proposed by other groups.
  4. The innovation contribution of this article is not clearly stated. The research contributions should be highlighted in the revised manuscript. There is a certain lack of a clear line and message, and my strong advice to the authors would be to consider the overall structure and to either significantly shorten the manuscript. We thank the reviewer for pointing out this. The amount of textual-based information stored electronically is rapidly increasing. Accumulating data is easy; finding relevant information on demand can be difficult as the size of the collection continues to escalate. This article presents a general framework for text mining consisting of a large body of EHRs with text data that are inherently unstructured and fuzzy (Pedianet database) and cutting-edge advances in deep learning. The derived variable (i.e., VZV infection yes/no) is then used to estimate VZV incidence within Pedianet. The availability of a gold standard allowed us to evaluate the system performance and the bias introduced in the incidence estimation. We clarified the issue in the text.
  5. Please explain and define all the variables in the equations and check the manuscript thoroughly and define the variables where necessary, otherwise, readers cannot understand the equations. We thank the reviewer for the suggestion. The original text does not report any explicit equation; we explicitly write the relevant ones, defining all the variables used in the text.
  6. There are many linguistic and grammatical typos. please carefully read through and conduct the proofreading. We checked the whole manuscript with Grammarly (including pro linguistic features). Additionally, we have required the MDPI language revision (editing certificate will be available upon request).

Round 2

Reviewer 2 Report

The authors have incorporated all suggested comments and now it can be accepted for publication.